# Position: The Most Expensive Part of an LLM *should* be its Training Data

**Nikhil Kandpal** [1]   **Colin Raffel** [1]

## Abstract

Training a state-of-the-art Large Language Model (LLM) is an increasingly expensive endeavor due to growing computational, hardware, energy, and engineering demands. Yet, an often-overlooked (and seldom paid) expense is the human labor behind these models' training data. Every LLM is built on an unfathomable amount of human effort: trillions of carefully written words sourced from books, academic papers, codebases, social media, and more. This position paper aims to assign a monetary value to this labor and argues that the most expensive part of producing an LLM *should* be the compensation provided to training data producers for their work. To support this position, we study 64 LLMs released between 2016 and 2024, estimating what it would cost to pay people to produce their training datasets from scratch. Even under highly conservative estimates of wage rates, the costs of these models' training datasets are 10-1000 times larger than the costs to train the models themselves, representing a significant financial liability for LLM providers. In the face of the massive gap between the value of training data and the lack of compensation for its creation, we highlight and discuss research directions that could enable fairer practices in the future.

## 1. Introduction

Investment into Large Language Models (LLMs) has recently surged, inspired by the widespread impact of AI systems and their anticipated profitability in coming years. A major driver of this investment is the reliable performance gain that comes from scaling up training runs (Kaplan et al., 2020; Hoffmann et al., 2022), making each generation of LLMs more expensive than the last. In fact, the cost of training a state-of-the-art LLM is currently doubling every

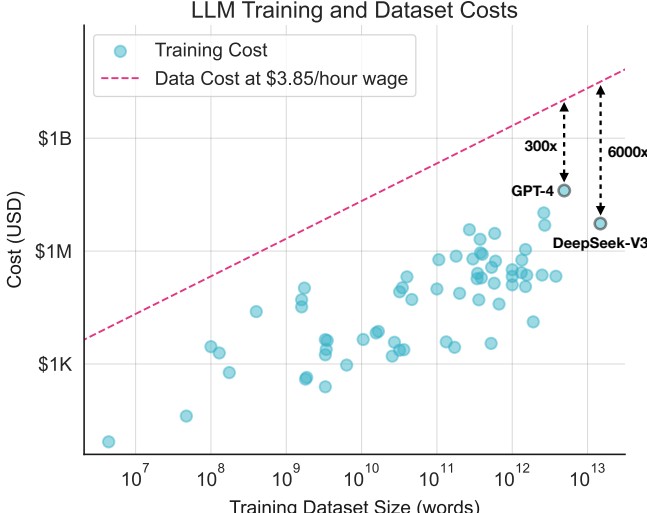

*Figure 1.* Estimated costs for LLMs' training datasets surpass the costs of training by 1-3 orders of magnitude. Above, we plot the training costs of 64 language models released between 2016 and 2024 along with the estimated cost of their training datasets.

nine months, with recent models like OpenAI's GPT-4 and Google's Gemini Ultra estimated to have cost tens of millions of US dollars (Cottier et al., 2024) and billion-dollar training runs projected in the near future (Morales, 2024).

Amid these rapidly escalating costs, one major expense remains largely unaccounted for: the large-scale text datasets used for training. Unlike other resources needed to produce LLMs, like hardware or energy, most training data has historically been collected *for virtually no cost* by mining text from the public Internet (Common Crawl). This web-scraped data is foundational to LLMs, particularly in their pre-training phase, where models require vast amounts of text to learn general linguistic and world knowledge. Notably, we are not aware of any compensation being given to the creators of this web text, despite their pivotal role in the success of modern LLMs.

In recent years, the practice of training LLMs on web text has come under scrutiny for both legal and ethical reasons. Much of the data on the Internet is protected under copyright law, making its use for training commercial LLMs the subject of multiple ongoing lawsuits (New York Times, 2023;

---

[1]University of Toronto & Vector Institute. Correspondence to: Nikhil Kandpal <nkandpa2@gmail.com>.

*Proceedings of the 42nd International Conference on Machine Learning*, Vancouver, Canada. PMLR 267, 2025. Copyright 2025 by the author(s).

Authors Guild, 2023; Concord Music Group, 2024; Alden Newspapers, 2024). In addition to being a legal gray area, training on authors' works without their consent has been argued to be exploitative, as language models possess the capability to outcompete the same creators whose data they were trained on (Pasquale & Sun, 2024). For these reasons, many corporate entities like publishers (The Atlantic, 2024; Reuters, 2024; Wiley, 2024; Vox Media, 2024) and social media platforms (Reddit, 2024; Shutterstock, 2023) have entered into data licensing agreements with LLM providers. Yet, even with these large players receiving compensation for their data, the vast majority of LLM training data comes from uncompensated authors.

Given this state of affairs, a natural question is how much compensation is owed to the training data contributors whose efforts underpin modern LLMs. To explore this question, we derive a deliberately low estimate of the labor costs required to create modern training datasets, based on the shortest feasible time and minimum wages needed for workers to write an equivalent volume of text. Even using this conservative estimate, we find that the cost of producing training datasets would be orders of magnitude larger than *all* other costs associated with training combined, such as hardware, energy consumption, and R&D staff salaries. **Thus, we claim that while LLM training is capital-intensive, the most expensive part of producing an LLM *should* be compensating the people that create its training data**.

This insight raises a new question of whether LLM providers are even financially capable of fairly compensating the labor involved in creating their training datasets. Our analysis shows that even at the modest wage rates assumed in our estimate, compensating data contributors is impractical for all but the wealthiest organizations. On its own, state-of-the-art LLM training is an expensive endeavor accessible to only a handful of companies, and when factoring in the costs of training data, the financial barriers of developing frontier language models become even more prohibitive.

While compensating training data contributors presents both practical and financial challenges, we conclude by highlighting promising research directions that could help make fairer compensation viable. In particular, we discuss the importance of adapting our current data collection practices, developing algorithms to efficiently balance the cost of training and data, and studying the benefits and drawbacks of different compensation structures. Progress in these areas can help pave the way to addressing the glaring lack of compensation provided to the individuals whose data makes these models possible.

## 2. Estimating LLM Development Costs

To support our claim that the hidden cost of training data dramatically outweighs other LLM development costs (such as hardware, energy, or engineering labor), we begin by defining principled methods for estimating each of these costs. This section details our methods for calculating the total cost of training a given LLM (Section 2.1), as well as the labor cost associated with producing a particular LLM's training dataset (Section 2.2).

### 2.1. Estimating Training Costs

We estimate the total cost to train models based on the methodology developed in Cottier et al. (2024), who analyze scaling trends in AI between 1950 and 2024. Cottier et al. (2024) estimate training costs based on variety of factors such as the cost of the hardware used for training, the energy consumption of that hardware, and the R&D staff salaries at the institution where the model was trained.

Specifically, Cottier et al. (2024) start with a comprehensive database of 890 notable AI models from Epoch AI (2024). For a subset of these models, they document the training hardware and estimate this hardware's cost by its retail value adjusted for depreciation between its release date and the time of the training run. They then estimate the energy consumption of the training run using historical energy prices and power consumption characteristics of the training hardware. Finally, for a small set of models, they factor in the cost of engineering labor based on publicly available salary data for the company conducting the training run and the number of contributors named in the model's technical report. The total training cost reported by Cottier et al. (2024) is the sum of these hardware, energy, and (optionally) engineering costs.

For our analysis, we adopt this pricing strategy, specifically focusing on models that operate in the language domain (e.g., language models, vision-language models, etc.). Furthermore, we only consider models studied by Cottier et al. (2024) for which the training dataset size is known, as we wish to ultimately compare LLM training costs with the costs to produce LLM training datasets.

### 2.2. Estimating Dataset Production Costs

The value of an intangible good, like training data, can be estimated in a variety of ways, each grounded in a different economic tradition. From the perspective of the labor theory of value (Marx, 1867), the worth of the data corresponds to the cumulative human effort involved in its creation. In contrast, the subjective theory of value (Menger, 1871) holds that value is determined by the preferences and judgments of those who use the data, meaning its worth arises from how much someone is willing to pay for access, regardless

| Writing Medium | Typical Length (words) | Estimated Writing Time | Estimated Cost (USD) |
|---|---|---|---|
| Blog Post | 2,000 | 1.1 hours | $4 |
| Academic Paper | 5,000 | 2.8 hours | $11 |
| Novel | 70,000 | 1.6 days | $150 |
| Textbook | 300,000 | 1 week | $642 |
| Encyclopedia Britannica | 40,000,000 | 2.5 years | $85,560 |

*Table 1.* Estimated writing times and costs for a variety of common text mediums based on our *extremely conservative* estimation approach. We aim to intentionally *underestimate* total costs to ensure claims about training data's relatively high costs are robust, even though real-world costs might be much higher (for example, the cost of creating the Encyclopedia Brittanica was about $400\times$ higher than our estimate).

of how it was produced. A third view emphasizes market-based valuation (Marshall, 1890), where value is dictated by the balance of supply and demand: rare or high-quality data with few substitutes is valued highly, while widely available or redundant content is worth less. Each of these approaches offers a different lens through which to evaluate the economic significance of pre-training corpora. In this analysis, where our goal is to estimate the compensation owed to training data creators, we take a labor-based approach that estimates the value of a training dataset by its *replacement cost*: the total labor cost associated with recreating an equally useful training dataset from scratch.

Modern large-scale text datasets include content from a wide variety of sources, including educational text, literary prose, marketing copy, social media content, and more. Estimating the labor costs to produce such a heterogeneous collection of text is difficult, as the expertise and time needed to produce different kinds of text vary greatly. Furthermore, most LLM providers only disclose the size of their training data rather than its content, making it impossible to know the prevalence of the different types of text in a given model's training set. To simplify this problem, we ignore text quality and only take into account the (approximate) number of word units in a dataset when estimating its cost. With this simplification, our cost-based valuation strategy reduces to a single question: How much would it cost to pay workers to write a collection of coherent text equal in size to a given training dataset?

To answer this question, we focus on estimating the speed at which an average person can produce coherent text as well as an hourly wage that might be paid to training data writers. We intentionally bias our analysis towards conservatively underestimating the value of text creation so that we can be confident that claims about the relatively high cost of training data will hold up regardless of how costs are estimated.

Specifically, we first assume contributors write at 30 words per minute, which is on the lower end of typical typing speeds among average computer users. However, we em-

phasize that this typing speed assumes that virtually no time is taken to give thought to what should be written, and any text that requires consideration, research, or editing would likely be written at a much slower rate. To further contextualize our estimate, at our assumed writing speed it would take about two hours to write this paper.[1]

Choosing an appropriate hourly rate is even more fraught because of the huge range of labor costs across the world. One might assume that an LLM developer would aim to minimize costs and therefore seek out writers willing to work for the bare minimum rate. On the other hand, an LLM developer might prioritize text quality and find that paying a higher rate results in text that produces better language models. Concretely, current public estimates of data annotation rates for the full gamut of machine learning tasks range from around $1 to $50 USD per hour (Dzieza, 2023). In an attempt to find a middle ground between these extremes, we rely on the government-mandated minimum wages across 167 countries (International Labor Organization, 2025). Specifically, we assume an hourly wage of $3.85 USD per hour, corresponding to the median minimum wage across these countries. Again, we emphasize that this is an intentionally low estimate so that we can ensure our claims are robust.

To provide intuition for how our intentionally conservative approach values different quantities of text, we list the estimated writing times and costs of common text sources in Table 1. As intended, these choices of writing speed and wages yield estimates that almost certainly underestimate the market value of text, particularly for high-quality sources like textbooks or encyclopedias. For instance, this valuation approach estimates the cost to write the Encyclopedia Britannica at $85,560 USD, while in reality the investment required to produce the encyclopedia's 15th edition, excluding printing costs, was $32 million USD (Encyclopedia Britannica)–about $400\times$ higher than our estimate. Therefore, any claims we make could likely be made more extreme with more realistic or precise estimates.

---

[1]It took considerably longer.

## 3. Cost Analysis Results

Using the valuation techniques described in Section 2, we estimate the training and dataset costs of 64 language models released between 2016 and 2024. In this section, we highlight some takeaways from our cost analysis.

### 3.1. LLM Dataset Costs Outweigh Training Costs

Despite our conservative approach to quantifying training data labor costs, the cost of data significantly exceeds training costs for every model we study. Shown in Figure 1, training data costs are at least an order of magnitude larger than training costs, with the costs of datasets for models like GPT-4 and DeepSeek-V3 being $300\times$ and $6000\times$ larger than the costs of their training runs respectively. This disparity underscores the relatively massive value of LLM training data, a cost which is almost never paid out to data creators.

We again note that more realistic estimates would make this disparity even greater. For example, if training data curation for language models had a comparable per-word cost to the creation of the Encyclopedia Brittanica, our data cost estimates would be multiple orders of magnitude higher. Notably, recent state-of-the-art language models are trained on increasingly high-quality and educational text (Penedo et al., 2024; Abdin et al., 2024a;b), suggesting that these higher hypothetical costs may in fact be more realistic.

### 3.2. Few Companies Can Afford Large-Scale Datasets

While many recent models rely on datasets that would be valued at over $10 billion USD by our conservative estimates (Figure 2), LLM providers today pay little to no money for training data, as most text is freely scraped from the Internet. Thus, we assess whether today's major LLM providers could afford to appropriately compensate the creators of their training datasets at the modest rates assumed by our data valuation model.

To quantify this, we consider ten organizations that have recently trained LLMs on large-scale datasets and compare the estimated creation cost of each company's training dataset to their annual revenue (see Figure 3). For 8 out of 10 of these companies, the cost of their training data exceeds 10% of their annual revenue, representing a substantial financial burden. For 3 out of 10, data costs surpass the company's total annual revenue, which would make it financially infeasible for them to compensate training data contributors even at the very low rates that we consider. These findings suggest that if companies were required to compensate the creators of their training data, even at the lowest possible rates, only the wealthiest companies could still afford to train state-of-the-art LLMs.

### 3.3. Training Dataset Costs Will Continue to Grow

The amount of uncompensated human labor underlying large-scale training datasets is already immense, but this debt can be expected to grow in the coming years. Driven by neural scaling laws (Kaplan et al., 2020; Hoffmann et al., 2022), LLM providers have historically expanded both model and dataset sizes to improve the performance of their LLMs. Moreover, recent trends indicate a growing preference for scaling training data over model size (Touvron et al., 2023b;a; Gadre et al., 2024). This shift is largely due to cost considerations: scaling up datasets is virtually free when scraping web text and the higher training costs incurred by training on more data are amortized against the lower cost of deploying a smaller model. As a result, training datasets – and their implicit monetary value – are growing rapidly, doubling every eight months according to Epoch AI (2024). Given that current LLM training datasets represent only a small fraction of the available text indexed on the Internet (Villalobos et al., 2024), this growth is likely to sustain in the near-term as LLM providers continue to scale up data collection efforts.

Given this trajectory, the AI community should work toward establishing clear norms and best practices for training data compensation as soon as possible. Doing so will help ensure distribution of value across all parties that contribute to the success of AI systems.

## 4. Future Research Directions

Our analysis in Section 3 surfaces a tension in LLM training: while AI systems rely on billions of dollars worth of uncompensated labor, paying data contributors even at conservatively low rates would make LLM training at current scales infeasible for all but the most resource-rich organizations. Moreover, even for organizations with the resources to pay for training data, practical challenges remain in determining *who* to compensate and how to structure these payments.

In this section, we identify future research directions that, if further studied, could help enable fairer and more practical training data compensation. These include adapting data collection practices, creating training algorithms that make more efficient use of data, and designing compensation models that balance fair payment with financial sustainability for LLM providers.

### 4.1. Training Data Collection

**Permissively Licensed and Public Domain Text**  Text released into the public domain or with a permissive license, such as the Creative Commons Zero (CC0), Creative Commons Attribution (CC-BY), MIT, or BSD licenses, represents an alternative to training on copyrighted web text.

Unlike general web text, which is automatically protected against reuse and redistribution by copyright law, public domain and permissively licensed text is explicitly released by its creators with fewer or no restrictions on how it is allowed to be used. Since the creators of this text have granted broad permissions for its downstream use, it can be argued that these licenses lessen the obligation to compensate creators for training on their data.

Past work has demonstrated the feasibility of collecting public domain and permissively licensed text at scale, resulting in text corpora like the Open License Corpus (Min et al., 2024), Common Corpus (Langlais, 2024), and Common Pile (com, 2025), each containing hundreds of billions to trillions of words. A high-impact line of future work would be to expand these corpora, as each year the amount of public domain and permissively licensed text grows, and many sources of this text (such as historical or spoken text) remain untapped. Furthermore, future research should focus on how to best use this data to produce useful LLMs, as its distribution and domain coverage differ drastically from typical Internet text (Min et al., 2024).

**Opt-In Data Collection** Current web text collection relies on indiscriminate scraping from the Internet, with the only formal opt-out mechanism being the robots.txt exclusion protocol. This system assumes implicit consent for scraping (and subsequent training) unless a website owner explicitly opts out. Over time, an increasing number of web domains have used this mechanism to block scraping by major LLM providers (Longpre et al., 2024), yet evidence suggests that these requests are routinely ignored (Paul, 2024), highlighting the system's limitations. Moreover, if LLM providers were to compensate data contributors, this paradigm would need to shift, as any transaction requires active and enforceable consent from all parties.

A key research direction is exploring the feasibility of opt-in data collection at the scale required for LLM training Some recent examples of opt-in dataset initiatives include OpenAssistant Conversations (Köpf et al., 2023) and WildChat (Zhao et al., 2024), where users consent to the collection of their LLM chat data, as well as the Mozilla Common Voice platform (Ardila et al., 2020), where users provide and annotate speech samples.

However, these and other crowdsourced datasets differ fundamentally from opt-in web text collection. In such initiatives, users opt in to generating *new* data in some restricted domain — such as multi-turn conversations or speech samples — with prior knowledge that this data will be used in a training dataset. By contrast, an opt-in web text collection system would need to scale to the vast diversity of existing LLM training datasets. This would likely require users to retroactively share their *existing* web content, despite it not

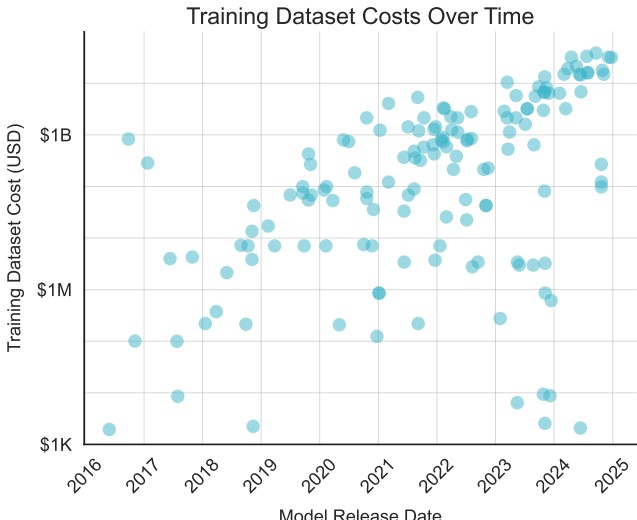

*Figure 2.* Over time the estimated labor costs to produce the content of LLM training datasets has increased, with numerous recent models having been trained on datasets that we conservatively estimate to have implicitly cost over $10 billion USD.

being originally created for inclusion in a training dataset.

**Authorship Provenance** When scraping data from an Internet webpage, many pieces of metadata, such as the web domain, URL, and the time of access, are typically collected and persisted alongside the page's text. However, one key piece of information not widely considered is *who* wrote the text on a particular webpage. While this information could in principle be collected if it was readily extractable, machine-readable authorship metadata is *not* available for most webpages.

There are existing mechanisms for establishing authorship of a webpage. One such mechanism is embedding the name of the author in a webpage's metadata via the <meta> element. The HTML standard (W3C, 2025) defines one use of this element as providing authorship information for a document. However, this tag is not widely used for this purpose as author information has little practical impact on, for example, search engine indexing and optimization. Even if it were, specifying only the name of an author is insufficient for the purpose of training data compensation, which would additionally require a means of directing payment to a specific individual.

To make training data compensation practical for web-scraped data, more thorough standards around how authorship information is presented on the Internet need to be established. Future work should look to design mechanisms for verified authorship of web text. Furthermore, collection and provenance of this metadata would need to be establish as a standard practice when collecting training data.

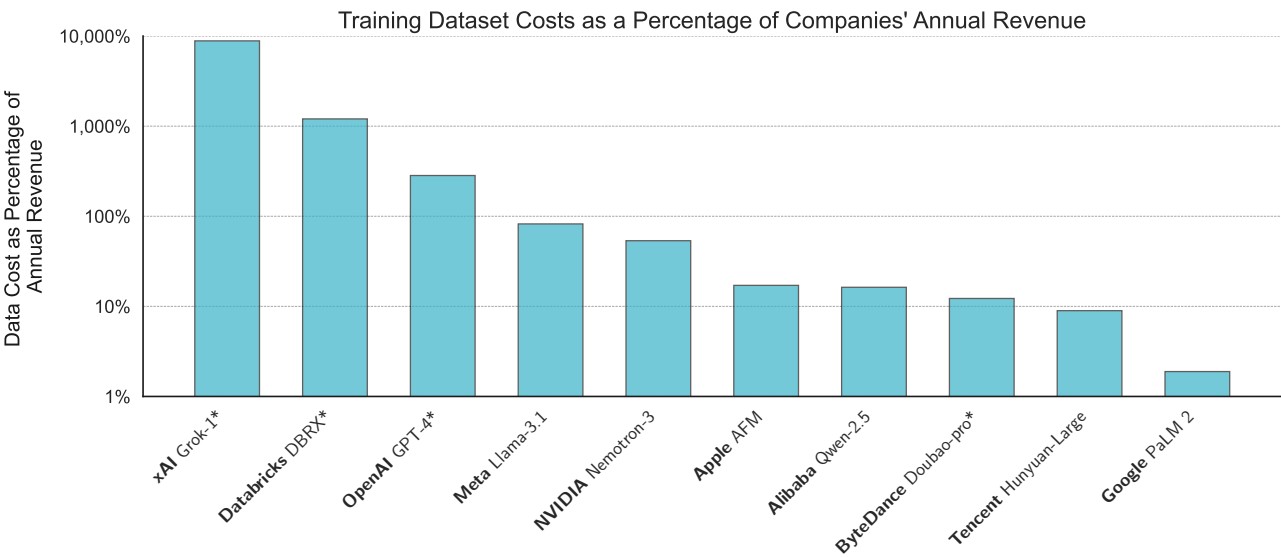

*Figure 3.* The costs of training datasets used by major LLM companies make up a significant fraction of these companies' revenues. Above we visualize training data cost as a percentage of each company's annual revenue for ten recent LLM training runs. We denote non-publicly traded companies denoted with * to indicate that for these companies we use third-party reported revenue rather than revenues reported in official financial filings.

## 4.2. Balancing Compute and Data Costs

**Data Efficiency** LLMs are considered to be data-inefficient, requiring trillions of tokens of training data in order to achieve strong performance. This inefficiency is the key driver of the immense labor costs that would be required to create modern training datasets. Future research should therefore focus on improving learning with smaller amounts of data.

Prior work has explored several approaches to improving the data efficiency of language models. For instance, Muennighoff et al. (2023) investigated how many times data can be repeated during training, finding that training an LLM on data repeated four times during training incurs little loss in quality. Other works have studied heuristics for filtering datasets down to relatively small, high-quality subsets that yield performant trained models (Penedo et al., 2024; Soldaini et al., 2024; Gadre et al., 2023; Xu et al., 2024). An orthogonal line of work studies the data efficiency of learning algorithms. One notable work in this area is the BabyLM Challenge (Warstadt et al., 2023) where participants compete to train the best possible language model on a limited-size dataset.

**Price-Optimal Language Models** Research on neural scaling laws (Kaplan et al., 2020; Hoffmann et al., 2022) has characterized the relationship between a language model's parameter count, training dataset size, and language modeling performance. This line of work has been highly influential, enabling LLM providers to determine the the optimal balance between model and training dataset size for a fixed compute budget.

Given the immense estimated costs of modern training datasets, we suggest expanding this framework beyond compute-efficiency to incorporate monetary constraints. Instead of solely optimizing for model performance given a fixed compute budget, future research should explore the best way to allocate a fixed financial budget between acquiring training data and training an LLM on the resulting training dataset. Analyzing the performance of price-optimal LLMs that take into consideration cost could provide insights into the tradeoffs between cost and model quality, helping the community understand what performance levels are achievable when training data costs are explicitly accounted for.

## 4.3. Training Data Compensation Structures

**Variable Payouts via Data Valuation** Text in large-scale datasets naturally exhibits a large amount of variation, with some data being higher-quality, and ultimately more useful for important downstream tasks (Penedo et al., 2024; Li et al., 2024). This variance in quality suggests that it might be reasonable to consider differing levels of compensation depending on some notion of text value.

The field of data valuation studies methods for quantifying how valuable individual training samples are to a model's predictions at inference time. For generative models like LLMs, value is usually formalized as the amount by which

a model generation decreases in likelihood when a particular training sample is removed from the model's training dataset (Choe et al., 2024; Park et al., 2023). Estimating the value of each training sample in a large training dataset is generally computationally intractable, since, by definition, measuring a sample's value requires training a new model from scratch. Thus, much of the work in this area focuses on tractable approximations to this notion of value, using approaches such as first-order Taylor series approximations (Koh & Liang, 2020; Choe et al., 2024; Park et al., 2023), changes in model behavior between checkpoints (Pruthi et al., 2020), and ideas from cooperative game theory (Ghorbani & Zou, 2019).

Despite significant progress, many of these methods remain computationally impractical at LLM scale. While there exist cheaper model-agnostic methods based solely on the text similarity between training samples and a model's outputs (Hanawa et al., 2021), they are likely not accurate enough to enable confident and reliable attribution for high-stakes applications like determining compensation. Future research should focus on developing scalable data valuation techniques that can be applied efficiently to both large models and massive training datasets, making variable compensation for training data feasible and reliable in practice.

**Sustainable Compensation Structures**   Paying data creators upfront for their contributions is prohibitively expensive, as shown in Section 3. Even at modest rates, only the wealthiest companies could afford to compensate training data creators at scale, leaving smaller entities unable to participate in fair data compensation practices.

An alternative approach is to shift from direct payment to more flexible compensation structures. One example of such a system is a royalty-based model, where contributors receive a percentage of the revenue generated by the AI system they help train. This compensation model lowers the immediate financial burden on smaller organizations by distributing payments over time rather than requiring an unaffordable lump sum. While the total royalties over a model's lifetime may still be lower than the upfront payments larger companies can afford, this system offers unlimited upside for data contributors — if an LLM becomes highly profitable, contributors could earn significantly more than they would under a fixed-price compensation scheme.

Future research should explore compensation frameworks like this that provide data contributors with value beyond immediate payment, such as revenue-sharing models or other mechanisms that align incentives between model trainers and data providers.

## 5. Alternative Views

The main position of this paper is that an enormous amount of human labor goes into creating the content of modern LLM training datasets, and in view of this effort LLM providers should compensate these workers when using their creations to build lucrative commercial products. In this section, we discuss the main alternative views that contrast this perspective.

**LLMs as Transformative Systems**   One opposing perspective is that LLM providers are doing more than simply collecting and repackaging text, but rather transforming it into something fundamentally new, i.e. a system capable of generating novel content beyond what is expressed in its training data. Under this view, training data is analogous to the educational materials that humans learn from over the course of their lifetimes, and a person learning from publicly available resources generally does not need to pay the full costs of creating those resources.

For instance, consider the following example: A software engineer reads an informative blog post about the Rust programming language and, as a result, is able to make their company's software products run significantly faster. This engineer's impactful contribution to the company's success is recognized with a promotion and raise. In this situation, it would be unreasonable to argue that the software engineer *owes* the author of the blog post compensation, even though the knowledge gained from reading the blog post was directly responsible for their increased earnings and career advancement. Rather, this is viewed as a situation where the software engineer learned foundational skills and subsequently applied those skills in a new and unique way.

When applied to LLMs, this reasoning aligns with the fair use doctrine, which asserts that training AI models on copyrighted text is a transformative act rather than mere reproduction. Under U.S. copyright law, fair use allows limited use of copyrighted material without permission of the copyright holder. Proponents argue that since LLMs generate new content by synthesizing patterns learned from vast amounts of text, their training process qualifies as transformative. However, this legal question remains unsettled, with multiple ongoing court cases seeking to determine whether LLM training constitutes fair use (New York Times, 2023; Authors Guild, 2023; Concord Music Group, 2024).

**Amortization of Data Costs**   Another common view is that while the creation cost of pre-training data is enormous, this cost is incurred only once and could be amortized over many pre-training runs. In this view, data represents an upfront investment similar to infrastructure: once acquired, it can be reused repeatedly to train multiple generations of models. In contrast, training costs like energy and engineer-

ing labor scale with the number of training runs, and thus these would be the cost centers that dominate long-term budgets.

However, this reasoning only holds if data costs are of similar magnitude to training costs—which our analysis suggests is not the case. We estimate that the labor involved in producing the content of large-scale pre-training corpora would translate to a cost that is often orders of magnitude (10-1000×) larger than that of the hardware, energy, and labor needed to perform a single training run. As a result, the number of repeated training runs required to amortize the data cost enough for training compute to dominate the overall budget is extraordinarily large. In practical terms, few organizations retrain hundreds or thousands of times on the same data, meaning the data cost would remain a dominant—and largely unacknowledged—component of LLM development.

**Existing Compensation Structures for Web Text**  A third perspective contends that many pre-training data creators are already compensated for their work through the existing economic structures supporting the modern Internet. Much of the content in large-scale datasets originates from blogs, forums, news outlets, and other sites that monetize traffic via ads, subscriptions, or sponsored content. From this standpoint, the creators have voluntarily published content in publicly accessible spaces to benefit from the public Internet's monetization opportunities.

However, this argument does not meaningfully address the legal or ethical questions at the heart of large-scale data use. The fact that a creator received compensation for publishing their work in one context—e.g., through ad revenue—does not imply that they have consented to its repurposing for a fundamentally different use, such as training a commercial language model. That labor was originally compensated based on its expected use (e.g., individual human readership), not on the work being ingested into a system that internalizes its patterns and scales to millions or billions of downstream model queries. Furthermore, compensation through ad revenue or subscriptions does not override intellectual property rights, although whether data use for AI training is infringing under U.S. copyright law is still a question being debated in courts.

**Concerns About Innovation and Access**  Finally, opponents of training data compensation might fear that requiring compensation for training data could stifle AI innovation and reinforce existing power imbalances. If LLM companies were *forced* to pay for every piece of training data, only the wealthiest organizations would be able to afford large-scale training, creating barriers for smaller research groups and startups. In this way, strictly enforced compensation might reinforce monopolies rather than democratize AI.

## 6. Conclusion

The success of the AI industry as we know it today is built on the collective effort of countless individuals, who together have written an unimaginably large volume of text across nearly all conceivable topics. Every dataset used to train modern LLMs is a product of this immense human labor, nearly always used without the explicit consent or compensation of its creators. Commercializing this vast amount of effort without addressing these concerns constitutes a significant ethical, and potentially legal, issue.

The main purpose of this paper is to quantify what it would take to fairly compensate training data contributors, and the answer is clear: a staggeringly large amount of money. Even under conservative assumptions, the costs of proper compensation far exceed those of compute, energy, and engineering combined. This realization should serve as a wake-up call for the AI community: our current data practices are unsustainable, both ethically and financially.

Despite this, we do not believe that LLMs should be abandoned as an area of research and innovation. Rather we should rethink how we train and deploy LLMs, so as to ensure that the value generated by them is fairly distributed. The research directions discussed in this paper represent concrete steps towards this goal. By working towards more ethical and sustainable data practices, we can enable an AI-driven future that respects and rewards the contributions of those who make it possible.

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
