# OpenReview forum: "Position: The Most Expensive Part of an LLM *should* be its Training Data"
_ICML.cc/2025/Position_Paper_Track — ICML 2025 Position Paper Track poster_

### Official Review · Reviewer_uoFe · 2025-03-08

**Significance:** 4
**Argument Clarity:** 4
**Rating:** 4
**Confidence:** 4

**Questions:**

- Figure 1: Do you use the constant hourly wage $3.85 USD for all years of 2016-2024? Would it change the conclusion if inflation and other changes to the hourly wage is included in the calculation?

**Discussion Potential:**

4

**Paper Summary:**

This paper states a clean position “The most expensive part of an LLM should be its training data”. Their goal is to assign a monetary value to the labour that produced the training data. Through one approach to calculating this monetary value, they argue it would be dominating compared to the cost of the hardware, energy, and engineering salary costs. They argue the training data collected at no cost by mining text from the web such as through Common Crawl, is crucial to train current LLMs. As these models are now producing monetary value, the creators of the data used to train them should be compensated.

The paper mainly considers one approach to estimate the monetary value of the training dataset by calculating how much it would cost to pay workers to write a collection of coherent text from scratch equal in size to a given training dataset. The estimate is based a conservative hourly wage based on government-mandated minimum wages across 167 countries and a very conservative speed of writing coherent text at 30 words per minute. Based on this approach, the cost of recreating datasets used in many recent models is estimated at over $10 billion USD.

The paper continues by discussing the implications of their analysis. For example, at such costs, if companies were required to compensate the creators, only the wealthiest companies could still afford to train state-of-the-art LLMs. They further expand on future research directions. They discuss the potential for utilizing only permissively licensed data and public domain text, opt-in data collection, and proposal to report authorship metadata in pages for future compensation.

They also discuss other compensation structures that assign monetary value to data differently. For example, the variable payouts approach would assign value based on how much value the data provides after it’s used while a sustainable compensation structure shifts the compensation payment to a flexible compensation such as royalty-based models.

The paper finishes by discussing alternative views that oppose compensation for training data. For example, the fair use principle in copyright law permits the use of copyrighted material without explicit permission if the use is sufficiently transformative. There is still no legal precedent establishing LLMs use of training data as sufficiently transformative.

## Update after rebuttal
I thank authors for their response and suggest incorporating their response and further discussions on the related lawsuits and labor valuation to the paper. I maintain a positive rating of the work since it is valuable to spark discussion on the topic and presents a clear position.

**Position:**

Yes

**Position In Title:**

Yes

**Related Work:**

3

**Strengths And Weaknesses:**

Strengths:
The paper presents a well-supported position. The method for estimating the monetary value of data is well-defined and well-stated. The implications of this valuation have been thoroughly discussed. The opposing views have been expanded.

Weaknesses:
- There are various active lawsuits involving the collection and use of training data. The paper could improve by expanding on the arguments made in those lawsuits. Particularly, section 5 could benefit from an overall summary of active arguments and positions in these lawsuits (New York Times, 2023; Authors Guild, 2023; Concord Music Group, 2024).
- It is important to acknowledge labour valuation is a long-standing question in economics with diverse theories. The paper focuses on one specific data valuation model. This is a good approach to focus the scope of the paper and make a solid argument. However, the discussion on alternative value models could be expanded by references to the economic theories on labour valuation. For example, connections to Labor Theory of Value, Subjective Theory of Value, and Supply and Demand. The discussion does not have to be long.

Minor improvements:
- Fig 1: Could the y-axis be clarified to say “Data/Train Cost”? The points seem to only show the cost of their training while the line shows the data cost.
- Fig 1: Can there be more names corresponding to each point as well as a table that shows all the estimated costs for all models?

Typos:
- Line 089: “Our analysis in X”, X is missing
- Line 264: establish -> established
- Line 319: the the -> the

**Support:**

4

---

> ### Author Rebuttal · Authors · 2025-04-01
>
> Thank you for this thoughtful and well-reasoned review. Below, we summarize and address the weaknesses and questions identified in your review:
>
> *Weakness 1: Lack of discussion of lawsuits regarding training on copyrighted IP*
>
> This is a great suggestion. Currently we briefly mention the three lawsuits you mention in Section 5, but do not go into much detail about the arguments beyond stating that they are about whether LLM training is transformative and thus fair use. We will be adding more detail in Section 5 into the specific arguments put forward in these lawsuits.
>
> *Weakness 2: The labor valuation literature is diverse and the paper could use more discussion of different approaches to valuation.*
>
> Thank you for the excellent suggestion. We provide a brief and informal overview of these approaches at the beginning of Section 2.2. However, we agree that the paper would be enhanced by more discussion of these theories of valuation along with references to direct readers to the relevant literature. We will add additional discussion at the start of Section 2.2 to our updated draft.
>
> *Question 1: Is inflation factored in when estimating training dataset costs? If not, would this change the high-level takeaways of the analysis?*
>
> Our estimates of training data costs do not currently take into account inflation. However, this would be useful to include in our analysis as the global inflation rate between 2016 and 2024 was about 30%. We will factor this into our analysis and update Figures 1-3. However, this should not substantially change our takeaways as (1) the biggest reported pre-training datasets have largely been collected post-2023 and (2) a 30% decrease in wages does not change the overall message that training data would be extremely expensive to produce and would still outweigh the cost of training runs, given our analysis demonstrating that the cost of training data is many orders of magnitude larger than the cost of training.

---

> > ### Comment · Reviewer_uoFe · 2025-04-03
> >
> > I thank authors for their response and suggest incorporating their response and further discussions on the related lawsuits and labor valuation to the paper. I maintain a positive rating of the work since it is valuable to spark discussion on the topic and presents a clear position.

---

### Official Review · Reviewer_qpSK · 2025-03-13

**Significance:** 3
**Argument Clarity:** 3
**Rating:** 3
**Confidence:** 4

**Questions:**

In LLM post-train, data usually needs domain experts and is paid to produce. Additionally, the cost is easier to estimate – the data is usually not publicly available on the web thus its cost is not shared by other parties. Thus I suggest the authors focus more on the study of data cost for post-train, and how its quality affects LLM’s final performance.

**Discussion Potential:**

3

**Paper Summary:**

This paper claims that the most expensive part of producing LLM should be compensating the creators of the training data. The authors estimate the labor costs required to recreate modern training datasets and compare these costs to the actual costs of training LLMs. Their analysis indicates that the hypothetical costs of creating training datasets are significantly larger (1-3 orders of magnitude) than the costs of training the models.

## update after rebuttal

Most of my concerns have been addressed. Given the novelty of this work and the importance of the data for large model training, I have changed my score to "weak accept" accordingly.

**Position:**

No

**Position In Title:**

Yes

**Related Work:**

3

**Strengths And Weaknesses:**

Strengths
- This paper addresses LLM cost from a novel angle – the cost of data generation, which was rarely talked about in the community.
- To support the claim, this paper proposes comprehensive methods to estimate the cost of training LLM and producing the data.
- The analysis result is very surprising, that producing training data is significantly more expensive than training LLM, which is very different from what people observe in the real world.
- This paper is well written and easy to read.

Weaknesses
- The biggest problem of this paper is that it’s not like a position paper; instead, it raises a hypothesis (as in the paper title) and proposes methods to support its hypothesis, which is more like a non-position paper (though without technical experiments). After reading this paper, it’s unclear to me what the action items the authors would like the research community to do? What can be influenced in the research community?
- Another big problem is the estimation of the cost of data production. 1) In reality, the massive data on the web is not produced for model training, instead it’s usually published on media with its economic income, such as via ads. 2) Usually training data is generated only once (or very limited times), then used for training for many times (could be hundreds or even more). Thus it’s unfair to use the entire cost of data production to compare with model training, as the data cost should be evenly afforded by ads etc. and the number of model training.

**Support:**

2

---

> ### Author Rebuttal · Authors · 2025-04-01
>
> Thank you for taking the time to thoughtfully and thoroughly review our paper. Your feedback is greatly appreciated. Below, we summarize and address the questions and weaknesses identified in your review:
>
> *Weakness 1: This is not a position paper, but rather one that performs an analysis to validate the hypothesis that producing an LLM training dataset would be more expensive than LLM training.*
>
> It is true that our paper validates the hypothesis that training data would be extremely expensive to produce from scratch. However, the position of our paper is not “pre-training datasets *would* be expensive to produce”, but rather that “pre-training datasets *should* be expensive”. These are two fundamentally different claims.
>
> The first is a verifiable claim that is either true or false (given an agreed upon notion of “expensive”), and is supported by our analysis in Section 2. However, the second claim is less black-and-white as there are arguments for and against the idea that pre-training data contributors should be compensated at reasonable market rates. The main argument for compensating data contributors is that their labor, in aggregate, is immensely valuable as evidenced by our estimated production costs. However, others might argue, as we do in the first paragraph of Section 4, that forcing companies to pay for all of their training data would result in even fewer companies being able to train LLMs than there are today. We additionally highlight other alternative views in Section 5.
>
> Finally, the second half of our paper proposes a variety of research directions for the AI research community that can lead to fair compensation for data contributors while also avoiding stifling innovation by pricing nearly all companies out of LLM development. Ultimately, our goal is to spark conversation on this topic since this is an aspect of the LLM development process that is historically overlooked.
>
> *Weakness 2: Text published on the Internet typically earns revenue via ads, and the existence of this revenue stream makes compensation from LLM trainers unnecessary.*
>
> This is an interesting point that we believe merits discussion in our paper. However, (1) not all of the text on the web is monetized via ads and (2) the purpose of our argument is not to say that authors are not getting compensated at all, but rather they should not be uncompensated for *this particular* use of their work. That being said, we will certainly add this perspective in our discussion of alternative views in Section 5.
>
> *Weakness 3: Text is something that can be purchased once and then used in an arbitrary number of training runs. Thus, comparing one training run to the cost of one dataset is not fair.*
>
> Thank you for bringing up this excellent point. We will be adding this to Section 3 of our paper when discussing our comparison of training data vs. training run costs. We do note, however, that training runs for models like DeepSeek-V3 and GPT-4 are 2-3 orders of magnitude less expensive than their training datasets. While this gap is likely somewhat mitigated by the fact that training data can be reused across training runs, it is still likely the case that the cost of data would still outweigh the cost of training.
>
> *Question 1: Post-training data contributors are typically compensated and an interesting line of inquiry would be to study how post-training data quality effects model performance.*
>
> This is certainly an interesting and impactful area to study to enable more effective post-training. However, this is orthogonal to the subject of this position paper, which is whether pre-training data contributors, who by and large are uncompensated, should be paid for their work.

---

> > ### Comment · Reviewer_qpSK · 2025-04-03
> >
> > Thanks for the detailed explanation!
> >
> > > “pre-training datasets should be expensive”
> >
> > Could you please further clarify the following questions a bit:
> > - Yes, building datasets are expensive. But how expensive compared to other factors (also mentioned by another reviewer)? Any quantitative estimation?
> > - What would be the potential different results if we compensate data contributors or not? How will this affect the current AI community? Will compensating data contributor be practical in the real world? And how to push this (e.g. legally)?

---

> > > ### Author Response · Authors · 2025-04-08
> > >
> > > Thank you for taking the time to engage with our response! Below, we provide some clarification for your questions:
> > >
> > > *1. Yes, building datasets are expensive. But how expensive compared to other factors (also mentioned by another reviewer)? Any quantitative estimation?*
> > >
> > > Our paper is aimed precisely at addressing this question. In Section 2.2, we provide quantitative estimates of how much modern pre-training datasets would cost if data contributors were compensated at extremely conservative market rates. In Section 3, we then compare these estimated costs to other major expenses involved in building LLMs – such as compute, energy, and labor (i.e., salaries for researchers and engineers).
> > >
> > > Our main finding is that, even under conservative assumptions, the cost of training data would far exceed all other major costs. This is the central reason we argue that training data should be considered the most expensive part of the LLM development pipeline.
> > >
> > > *2. What would the potential different results be if we compensated data contributors? How would this affect the current AI community?*
> > >
> > > Compensating data contributors would have wide-ranging effects. As we discuss in Section 3.2, even under conservative wage estimates, such compensation would be prohibitively expensive for most companies. If required, it could limit the ability to pre-train models to only a small number of well-resourced organizations.
> > >
> > > However, taking data compensation seriously could also positively reshape research priorities in the field. In Section 4, we discuss how this shift could stimulate work on new research directions – including the curation of permissively licensed datasets, the development of more data-efficient algorithms, and the design of financially feasible compensation frameworks for data contributors.
> > >
> > > *4. Would compensating data contributors be practical?*
> > >
> > > In our paper, we address this question in two ways:
> > >
> > > - Affordability (Section 3.2): As noted, compensating contributors using our conservative cost model would represent a substantial burden, likely unaffordable for most companies at the scale of today’s pre-training datasets.
> > >
> > > - Feasibility of implementation (Section 4.1): Even if compensation were affordable, implementation presents significant logistical challenges. For example, current data collection practices largely rely on scraping public web data, which typically lacks reliable author attribution. As a result, it's often unclear who should be compensated for a given piece of content.
> > >
> > > That said, these challenges are largely a consequence of current norms – norms that evolved without data contributor compensation in mind. If the AI community begins to treat contributor compensation as a serious design consideration, we believe there are a number of ways to mitigate both cost and implementation barriers. In Section 4, we detail a number of promising research directions aimed at making compensation more practical in terms of both cost and implementation.
> > >
> > > *5. And how to push this (e.g., legally)?*
> > >
> > > Legal pressure may play a key role in shifting norms. As we briefly discuss in Sections 1 and 5, the legality of using scraped copyrighted content for training LLMs is currently under debate in several U.S. lawsuits. In response to your and Reviewer uoFe’s suggestions, we plan to expand our discussion of these ongoing legal arguments and their implications in the final version of the paper.

---

### Official Review · Reviewer_t5EP · 2025-03-14

**Significance:** 4
**Argument Clarity:** 4
**Rating:** 4
**Confidence:** 4

**Questions:**

- It seems that the paper is more in the legal and copyright view. If there is still no laws or regulations on this issue, it's less likely that the big tech companies will put any efforts on resolving this issue and giving compensation to the data creators.

**Discussion Potential:**

4

**Paper Summary:**

This paper argues that the hidden cost of modern large language models lies in the human labor required to produce their training data, not in the training process. By analyzing 64 LLMs released between 2016 and 2024, the authors use conservative assumptions (e.g., low-end typing speeds and median minimum wages) to show that the estimated costs for producing training datasets are one to three orders of magnitude higher than the actual training costs. The paper also propose to establishing opt-in data collection mechanisms, improving data efficiency, and designing merit-based schemes to compensate the data creators.

**Position:**

Yes

**Position In Title:**

Yes

**Related Work:**

4

**Strengths And Weaknesses:**

Strengths:

- The paper offers a fresh viewpoint by shifting the focus from traditional training costs to the hidden cost of creating the pretraining data. This viewpoint is seldom proposed by prior work. But it's a very practical issue, and the mechanism of how to compensate the data creators have to be addressed sometime in the future.

Weaknesses:

- Consider this analogy: when a lawyer studies using freely available online resources to acquire legal expertise, clients pay for the lawyer’s professional advice and their efforts in learning laws, rather than compensating for the educational materials the lawyer used. Similarly, for retrieval-augmented language models (RAG), if a generic LLM is deployed and clients supply their own specialized knowledge—such as in medical or legal fields—the clients should only be charged for the model’s training and service capabilities, not for the underlying cost of the knowledge sources. This similar point is mentioned in the section 5.
- Line 304-306 (left): MIssing citation of MetaCLIP [1] which filter the data to a minimum subset. The idea of maintaining "metadata" in the MetaCLIP can also benefit the merit-based compensating system this paper tried to propose.

[1] Demystifying CLIP Data https://arxiv.org/abs/2309.16671

**Support:**

3

---

> ### Author Rebuttal · Authors · 2025-04-01
>
> Thank you for taking the time to thoughtfully engage with our paper. Below is some discussion of the weaknesses and questions identified in your review.
>
> *Weakness 1: Human experts, e.g., lawyers, are paid for their expertise at applying learned concepts rather than compensated for the value of their educational materials.*
>
> The example you identified is certainly one of the main arguments against training data compensation – that LLMs learn from their training data similarly to how humans learn from publicly available materials, and LLM trainers, like human learners, should not be obliged to pay for publicly available educational resources.  This is highlighted as one of the alternative views we present in Section 5. This argument relies on the assumption that LLMs learn similarly, and should be treated similarly, to humans. There is some evidence against this perspective based on LLMs’ tendency to memorize individual training examples [1,2,3], at times at the expense of generalization [4]. However, whether legal precedent is established to support or oppose this viewpoint remains to be seen. Until that precedent is established, we argue that it is worth considering the human labor required to create LLM training datasets, which as of yet is largely unaccounted for.
>
> *Weakness 2: Should data introduced to LLMs via RAG be treated distinctly from training data?*
>
> We believe this is an interesting perspective, but somewhat orthogonal to our discussion of whether training data creators should be compensated. However, based on your suggestion, we will plan to add some discussion on this topic as there is already some existing literature on this problem of attributing a retrieval-augmented LLM’s predictions back to the retrieved sources in its context [5,6].
>
> *Weakness 3: MetaCLIP should be cited when discussing approaches to quality-filtering large-scale datasets*
>
> Thank you for directing us to this relevant prior work. This is another excellent example of how new filtering techniques can yield substantially smaller training datasets while retaining or even improving model performance. We will certainly be citing this work in Section 4.2.
>
> *Question 1: Will LLM providers invest in compensating training data creators without legal or regulatory pressure?*
>
> We agree with the point you make, that LLM providers are unlikely to change their ways without external pressure. This is a large part of why we chose to write this position paper – few people were considering the cost associated with training data and we believed that the community would benefit from discussion on this topic.
>
> Please let us know if we have addressed your questions and concerns.
>
> 1. Nicholas Carlini, Florian Tramer, Eric Wallace, Matthew Jagielski, Ariel Herbert-Voss, Katherine Lee, Adam Roberts, Tom Brown, Dawn Song, Ulfar Erlingsson, Alina Oprea, Colin Raffel. Extracting Training Data from Large Language Models. https://arxiv.org/abs/2012.07805
>
> 2. Milad Nasr, Nicholas Carlini, Jonathan Hayase, Matthew Jagielski, A. Feder Cooper, Daphne Ippolito, Christopher A. Choquette-Choo, Eric Wallace, Florian Tramèr, Katherine Lee. Scalable Extraction of Training Data from (Production) Language Models. https://arxiv.org/abs/2311.17035
>
> 3. Nicholas Carlini, Daphne Ippolito, Matthew Jagielski, Katherine Lee, Florian Tramer, Chiyuan Zhang. Quantifying Memorization Across Neural Language Models. https://arxiv.org/abs/2202.07646
>
>
> 4. Xinyi Wang, Antonis Antoniades, Yanai Elazar, Alfonso Amayuelas, Alon Albalak, Kexun Zhang, William Yang Wang. Generalization v.s. Memorization: Tracing Language Models' Capabilities Back to Pretraining Data. https://arxiv.org/abs/2407.14985
>
> 5. Benjamin Cohen-Wang, Harshay Shah, Kristian Georgiev, Aleksander Madry. ContextCite: Attributing Model Generation to Context. https://arxiv.org/abs/2409.00729
>
> 6. Fengyuan Liu, Nikhil Kandpal, Colin Raffel. AttriBoT: A Bag of Tricks for Efficiently Approximating Leave-One-Out Context Attribution. https://arxiv.org/abs/2411.15102

---

> > ### Comment · Reviewer_t5EP · 2025-04-03
> >
> > Thanks for the reply! The new information addressed my concerns and show the value of this position paper. I will raise my score for accept.

---

### Official Review · Reviewer_tsCD · 2025-03-19

**Significance:** 2
**Argument Clarity:** 2
**Rating:** 2
**Confidence:** 3

**Questions:**

How about compare training data costs with other costs such as costs of Infrastructure & Storage, Engineering & Research Talent.

**Discussion Potential:**

2

**Paper Summary:**

The paper claims that the most expensive part of producing an LLM should be the compensation provided to training data producers for their work.

**Position:**

No

**Position In Title:**

Yes

**Related Work:**

2

**Strengths And Weaknesses:**

Strength: The evidences are supportive.

Weakness: Lack of comparing between data cost and other costs, such as costs of Infrastructure & Storage, Engineering & Research Talent.

**Support:**

2

---

> ### Author Rebuttal · Authors · 2025-04-01
>
> Thank you for taking the time to review our position paper. Below we address the weaknesses raised in your review:
>
> *Weakness 1: Lack of comparison between data cost and other costs, such as the costs of infrastructure & storage, engineering & research talent, etc.*
>
> We agree that infrastructure, storage, engineering, and other costs are important to include, and in fact this comparison is actually the core of the analysis provided in the paper. In Section 2.1 we detail the methodology (developed by Cottier et al., 2024) for estimating the non-data costs associated with producing LLMs. These include the amortized cost of training hardware, energy required to perform the training run, and salaries for the engineers and researchers who contributed to the LLM.
>
> Then in Section 3.1, we compare these costs with the estimated production cost of large-scale training datasets and find that data costs greatly outweigh the aggregate cost of hardware, energy, and engineering salaries combined for a wide variety of models trained between 2016 and 2024. This insight forms the basis of the position of this paper – that training data should be the most expensive part of an LLM.
>
> Please let us know if this addresses your concern about our paper.

---

### Decision · Program_Chairs · 2025-04-29

**Decision:**

Accept (poster)

**Comment:**

The paper analyzed how much the LLM training data costs, compared with other costs and advocated that training data should be the most expensive part and that data creators should be compensated in a merit-based system.

The paper advocates for a clear and non-obvious position with 3 out of 4 reviewers voted to accept. The only reviewer voting reject raised one point that is already clearly addressed by the paper. Most reviewers found the position to be interesting and engaged in discussions on valuation methods, pretraining vs post-training data, effect of inflation, legal questions on if LLM is fair use and what is a plausible way forward if their proposals are adopted. The authors already anticipated some of these issues, demonstrating that a good amount of thought were put into their position. It would be also interesting to speculate on some market based mechanics for compensating future pretraining data given that much post-training data is already paid for, rather than mostly deferring to external legal pressure.